# The COVID-19 pandemic and food insecurity in households with children: A systematic review

Anna Williams[1], Nisreen A. Alwan[1,2,3], Elizabeth Taylor[2,4], Dianna Smith[2,4], Nida Ziauddeen[1,2] *

1 School of Primary Care, Population Sciences and Medical Education, Faculty of Medicine, University of Southampton, Southampton, United Kingdom, 2 NIHR Applied Research Collaboration Wessex, Southampton, United Kingdom, 3 University Hospital Southampton NHS Foundation Trust, Southampton, United Kingdom, 4 School of Geography and Environmental Science, University of Southampton, Southampton, United Kingdom

* Nida.Ziauddeen@soton.ac.uk

## Abstract

### Background

Food insecurity is defined as not having safe and regular access to nutritious food to meet basic needs. This review aimed to systematically examine the evidence analysing the impacts of the COVID-19 pandemic on food insecurity and diet quality in households with children <18 years in high-income countries.

### Methods

EMBASE, Cochrane Library, International Bibliography of Social Science, and Web of Science; and relevant sites for grey literature were searched on 01/09/2023. Observational studies published from 01/01/2020 until 31/08/2023 in English were included. Systematic reviews and conference abstracts were excluded. Studies with population from countries in the Organisation for Economic Co-Operation and Development were included. Studies were excluded if their population did not include households with children under 18 years. The National Heart, Lung, and Blood institute (NIH) tool for observational cohort and cross-sectional studies was used for quality assessment. The results are presented as a narrative review.

### Results

5,626 records were identified and 19 studies were included. Thirteen were cross-sectional, and six cohorts. Twelve studies were based in the USA, three in Canada, one each in Italy and Australia and two in the UK. Twelve studies reported that the COVID-19 pandemic worsened food insecurity in households with children. One study reported that very low food security had improved likely due to increase in benefits as part of responsive actions to the pandemic by the government.

**Data Availability Statement:** All data included in this article come from published papers.

**Funding:** The Wessex DIET study which this review formed part of is supported by the NIHR Applied Research Collaboration Wessex. The funder had no

role in study design, data collection and analysis, decision to publish, or preparation of the manuscript. ET and NZ is supported by NIHR Applied Research Collaboration Wessex. The views and opinions expressed in this protocol are those of the authors and do not necessarily reflect those of the NIHR or the Department of Health and Social Care. For the purpose of Open Access, the author has applied a Creative Commons Attribution (CC BY) licence to any Author Accepted Manuscript version arising.

**Competing interests:** The authors have declared that no competing interests exist.

## Conclusion

Although studies measured food insecurity using different tools, most showed that the pandemic worsened food security in households with children. Lack of diversity in recruited population groups and oversampling of high-risk groups leads to a non-representative sample limiting the generalisability. Food insecure families should be supported, and interventions targeting food insecurity should be developed to improve long-term health.

## Introduction

Lockdowns and other strategies to prevent the spread of SARSCoV2 led to disruptions in employment and increasing experience of adversities [1, 2]. Data from the Office for National Statistics showed that there were 220,000 fewer people in employment between April to June 2020 (first three months of lockdown) than between January and March 2020 in the UK [3]. Older and younger workers, part-time workers and self-employed were the worst affected [3].

The Food and Agriculture organisation of the United Nations describes a person to be food insecure when they "lack regular access to safe and nutritious food for normal growth and development and an active and healthy life" [4]. Global food insecurity has been increasing since 2014 (21.2%) but increased sharply (equivalent to the rise in the previous five years) in 2020 (29.5%) and remained high in 2021 [5]. While the prevalence of overall food security remained constant in 2021, the prevalence of severe food insecurity increased significantly implying that those previously facing moderate food insecurity were pushed into severe food insecurity. A total of 2.3 billion people faced food insecurity in 2021 [5]. In the UK, approximately 8% of the population were food insecure at this time, according to government estimates [6].

Adults with income losses resulting from COVID-19 measures were at increased risk of experiencing food insecurity than adults whose income had not been affected [7]. Households that were already food insecure experienced greater limits on diet quantity and quality during the pandemic, with negative impacts on physical and mental health or wellbeing [7–10]. Food insecurity is associated with lower dietary quality in adults but not consistently in children, potentially due to adults compromising their diet quality to shield children [11]. Food insecurity during early childhood can impair cognitive development due to poor nutrient intake. Exposure to increased anxiety and stress because of food insecurity could also impact development through physiological and psychological mechanisms. Children would also be less likely to take part in extracurricular activities due to low monetary resource further impacting their development [12].

To our knowledge, no systematic reviews have looked at rates of food insecurity in households with children before and during the COVID-19 pandemic in high income countries. We aimed to systematically review the current literature to describe the association between the COVID-19 pandemic and food insecurity in households with children (<18 years) from countries that are part of the Organisation for Economic Co-Operation and Development (OECD), used as a proxy for high income countries.

## Methods

The PICO (population, intervention, comparison, outcome) framework was used to develop a search strategy. The population was households with children under 18 years. The intervention

**Table 1. Inclusion and exclusion criteria for the review.**

| Inclusion criteria | Exclusion criteria |
|---|---|
| Case studies, cross-sectional and cohort studies | Systematic or other review articles, randomised controlled trials, mixed methods studies, dissertations, conference abstracts and qualitative studies. |
| Households (two-parent, lone parent or any primary caregiver) with at least one child from 0–18 years of age | Adult households with no children over 18 years of age |
| Populations in countries that are part of the Organisation for Economic Co-Operation and Development (OECD), used as a proxy for high income countries. | Low-and medium-income countries |
| Studies published between 01/01/2020 and 31/08/2023 (updated systematic search date) | |
| Published in English | |

was the direct and indirect impacts of the COVID-19 pandemic. The comparator was the prevalence of food insecurity before the COVID-19 pandemic, or comparison to a different geographical area. The primary outcome was household food insecurity. Secondary outcomes were poverty status, mental health, diet quality and weight status.

Inclusion and exclusion criteria are presented in Table 1. Observational studies that examined the impact of the COVID-19 pandemic on household food insecurity in households with children were included. Studies with population from countries in the OECD [13] were included. Studies published after 01/01/2020 until the search date and in English were included. The search was run on 31/08/2022 and updated to 31/08/2023 in September 2023. Systematic reviews and conference abstracts were excluded. Studies were excluded if their population did not include households with children under 18 years. The search strategy was developed with input from a research librarian and is presented in Table 2.

Four electronic databases were searched: EMBASE (via Ovid), Cochrane library, International Bibliography of Social Science and Web of Science. Snowball sampling was also undertaken and reference lists were hand searched for relevant articles to be included into the screening process. A search of the grey literature was undertaken on the following organisation websites: Food Foundation, Sustain, Christians against Poverty, Trussell trust, Nourish Scotland, Independent Food Aid Network, Evidence and Network on UK Household Food Insecurity (ENUF), Joseph Rowntree Foundation and Citizen's Advice. The review was not registered.

**Table 2. Search strategy.**

| |
|---|
| (OECD or Australi* or Austria* or Belgi* or Canad* or Chile* or Colombia* or Costa Rica* or Czech Republic or czech* or Denmark or Danes or danish or Estonia* or Finland or Finnish or France or French or German* or Greece or greek or Hungar* or Iceland* or Ireland or Irish or Israel* or Ital* or Japan* or Korea* or Latvia* or Lithuania* or Luxembourg* or Mexic* or Netherlands or dutch or New Zealand* or Norway or Norwegian* or Poland or polish or Portug* or Slovak republic or slovak* or Sloven* or Spain or Spanish or Sweden or Swedish or Switzerland or Swiss or Turkiye or turkey or Turkish or United Kingdom or UK or Engl* or Scot* or Wales or welsh or Northern Ireland or Great Britain or Britain or British or USA or United States of America or US or United States or American or (Organisation for Economic Cooperation and Development)) AND |
| (Food poverty or food insecurity or food or food insufficiency or nutrition* security or food deprivation or diet* or diet* quality or diet* adequacy or diet* intake or nutrition* or nutri* requirements or energy intake or macronutrient* or micronutrient* or vitamin* or mineral* or infant food or infant nutrition or calor* or hunger or food quality or food availability).mp. AND |
| (isolation or lockdown or coronavirus or SARS-CoV-2 or quarantine or restrictions).mp. AND |
| (child* or infant* or toddler* or baby or babies or school age* or newborn or preschool or pre school or nursery or famil* or lone parent or single parent or household* or young child* or primary caregiver or parent* or teen* or young adult* or young or under 18 or <18 or college or sixth form or under 12 or <12 or primary school or school or secondary school or high school or reception or families with children or adolescent* or young pe*).mp. |

The screening management software Rayyan [14] was used for the screening of titles and abstracts for eligibility. A 10% random sample of the titles and abstracts were screened independently by two reviewers (AW, NZ). The agreement between reviewers was 92% and one author then screened the remaining titles and abstracts (AW). Conflicting decisions were mediated by a third reviewer (NAA). Full-text screening was completed by AW and NZ. Any conflicts were discussed, and reviewers had an agreement rate of 100%. Data was extracted independently from all included studies by two reviewers (AW, NZ) except for articles included through the updated search which was done by NZ. Agreement rate for data extraction was 100% between the reviewers. Items extracted from studies included study design, study location, sample size, data collection period, population, ethnicity, child age, exposure/comparison group, outcome measurement tools and outcome data.

The quality of each study was independently assessed by two reviewers (AW, NZ), using the National Heart, Lung, and Blood institute (NIH) quality assessment tool for observational cohort and cross-sectional studies [15]. Any disagreements were resolved in discussion. The grey literature study was excluded from the quality assessment as it was not applicable to the quality assessment tool.

## Results

A PRISMA flow diagram was used to document the screening process (Fig 1) [16]. 5617 records were identified through the electronic database search, 775 of which were duplicates. Titles and abstracts of 4842 records were screened, of these 4804 records were excluded, leaving 38 records for full-text screening. An additional eight records were included in full-text

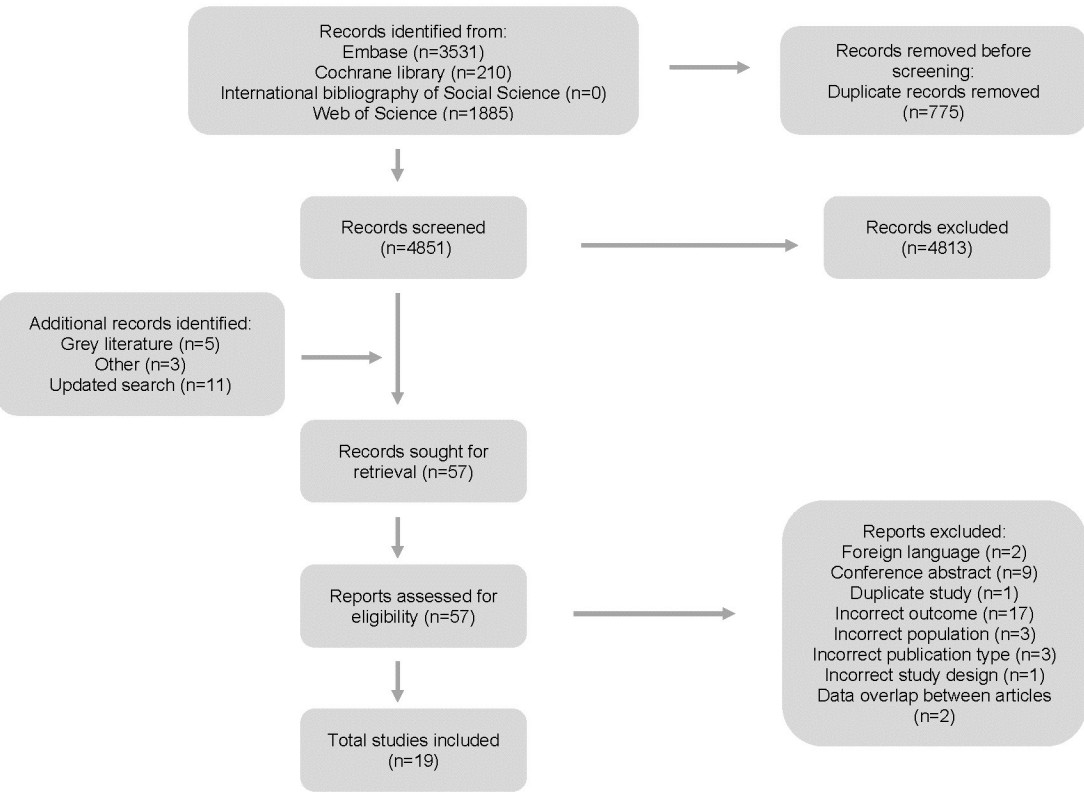

**Fig 1. PRISMA flow diagram.**

screening—five from grey literature searching, and three from reference searching. A total of 46 full-text records were screened and 15 were originally included in this narrative review with three articles being added from the updated search [17–35].

Study characteristics are described in Table 3. Thirteen studies were cross-sectional and six were cohorts. Of the 19 included studies, twelve studies were based in the USA [17–20, 25, 28, 29, 31–35], three in Canada [22, 23, 30], one each in Italy [24] and Australia [27], and two studies used data from both UK and Great Britain populations [21, 26].

The included studies used different measures for the outcome of food insecurity. Eight studies used the United States Department of Agriculture (USDA) food security survey modules [36], seven of which used the six-item [18–20, 27, 31–33] and one used the 18-item [25]. One study [26] captured moderate and severe experiences of food insecurity in adults using three modified items from the ten-item USDA Adult Food Security [36] and measured child food insecurity using four questions also used by the USDA. Six studies [17, 24, 28–30, 34] used the 2-item Hunger Vital Sign (HVS) [37]. One study [22] adapted the HVS question related to not having enough money to buy food to capture food security in the last month and over the next 6 months. One study [23] used Health Canada's 18-item Household Food Security Survey Module. One study [35] assessed food insecurity by asking a yes/no question about "worry about the amount or type of food available to you at home due to money or lack of availability". One study [21] asked one question each about quantity and access to sufficient food as proxy for food security based on the Food and Agricultural Organisation of the United Nations definition [4].

## Food Insecurity in households with children

All studies included food insecurity as an outcome of which thirteen studies compared the prevalence before and during the pandemic (Table 4, Fig 2). Twelve studies reported an increase in food insecurity during the COVID-19 pandemic [18–20, 23–25, 28–30, 32–34]. One of the twelve studies conducted three surveys during the course of the pandemic [18–20] and found that the prevalence of food security in the study population had returned to pre-pandemic levels but the prevalence of very low food security remained higher (9.6% pre-pandemic to 16.8% in May 2021) but had decreased from earlier pandemic survey time-points [20]. One study examined the prevalence of very low food insecurity in low-income households with children in the US [31] and reported a decrease of 5.3% from 2020 pre-COVID-19 restrictions and 8.2% from 2019. This decrease was likely due to an increase in nutritional assistance benefits as part of responsive actions to the pandemic by the government.

Three studies examined prevalence of food insecurity at different time-points during the pandemic [26, 27, 35] (Table 3). Two studies used repeated cross-sectional surveys, one at five time-points [26] in the UK beginning from the first two weeks of lockdown (March/April 2020) till January 2021 and the other [27] at three time-points in Australia beginning from May 2020 till May 2021. Both studies found that prevalence was highest at the first time-point early in the pandemic and decreased over time. The study in the UK [26] found that the prevalence in households with children was higher at all time-points than in households without children (20.8% compared to 13.7% in the first two weeks of lockdown and 9.6% compared to 6.6% in January 2021). The third study [35] recruited middle school students in the US and found that prevalence of food insecurity had increased from 28.4% in October 2020 to 30.3% in April/May 2021 however both time-points were analysed as cross-sectional data as only 56% (n = 49) responded to the survey at both time-points.

Three studies reported on food insecurity during the pandemic only [17, 21, 22]. Among 200 families screened for food insecurity during routine paediatric visits in April/May 2020,

**Table 3. Key characteristics of included studies.**

| Author, year | Study design | Location | Sample size | Data collection period | Population | Ethnicity | Child age | Exposure/ comparison | Quality assessment |
|---|---|---|---|---|---|---|---|---|---|
| Abrams et al [17] | Cross sectional | Austin, Texas, USA | 200 families | 14th April 2020–20th May 2020 | Parents and caregivers of paediatric patients (<2 years) | Hispanic 82% Non-Hispanic 17.5% Missing 0.5% | Child age <2 years | Before the COVID-19 pandemic | +sequential recruitment +use of validated screening tool for food security measurement -non-representative sample, predominantly of one ethnicity -COVID-19 restrictions and guidelines meant that sample not representative of usual age range seen at recruiting clinics |
| Adams et al 2020 [18] | Cross sectional | USA | 584 parents | 30th April–23rd May 2020 | Parents >18 years with at least 1 child 5–18 years old | Caucasian/ White 82.7% African American 6.0% Asian 4.3% American/ Indian 2.9% Other 6.5% | Mean child age 9.6 ± 3.8 years | Before COVID-19 pandemic | +reliable measure used to measure food security which also reduces participant burden -convenience sample -retrospective self-report for questions related to pre-COVID-19 |
| Adams et al 2021 [19] | Cohort (longitudinal survey) | USA | 433 parents | September 2020 | Parents >18 years with at least 1 child 5–18 years old | Caucasian/ White 84.8% African American 6.7% Asian 3.9% Other 6.7% | Mean child age 9.4 ± 3.8 years | Before COVID-19 pandemic and to earlier pandemic data collection in April/May 2020 | +high retention rate +reliable measure used to measure food security which also reduces participant burden -convenience sample -higher drop-out rate in participants from minority backgrounds and lower education |
| Adams et al 2023 [20] | Cohort (longitudinal survey) | USA | 333 parents | May 2021 | Parents >18 years with at least 1 child 5–18 years old | Caucasian/ White 84.7% | Mean child age 9.3 ± 3.7 years | Before COVID-19 pandemic and to earlier pandemic data collections in April/May 2020 and September 2020 | +57% retention rate at 1 year -convenience sample -higher drop-out rate in participants from minority backgrounds and lower education |
| Brown et al [21] | Cohort (data from Understanding Society) | UK | 9501 households | April, July and September 2020 | Households in the UK | - | 8% households with children aged 0–2, 6% 3–4, 17% 5–11, 13% 12–15 | Socioeconomic factors that may increase risk of food insecurity during the COVID-19 pandemic | +data from national longitudinal survey -unvalidated measure of food security -generalisability limited as analysis limited to those who responded to study questions of interest |

*(Continued)*

**Table 3.** (Continued)

| Author, year | Study design | Location | Sample size | Data collection period | Population | Ethnicity | Child age | Exposure/comparison | Quality assessment |
|---|---|---|---|---|---|---|---|---|---|
| Carroll et al [22] | Recruited from longitudinal cohort, analysis cross-sectional | Canada | 254 families (235 mothers (m), 126 fathers (f) and 310 children) | 20th April–15th May 2020 | Families with a child between 18 months and 5 years. Middle to high income families. | Caucasian 86.8% m, 88.1% f African American 0.9% m, 0 f Latin American 3% m, 2.4% f Asian 4.7% m, 4.0% f South/West Asian 3% m, 3.2% f Other 1.3% m, 0.8% f | Mean child age 6.0 ± 2.0 | Financial stress, stress and food insecurity assessed in the past month and over the next six months. | +recruited from longitudinal family based cohort +high response rate (83%) -only first parent to enroll was asked questions about participating children -responses from mothers and fathers from same household not differentiated in the analysis |
| Cyrenne-Dussault et al 2022 [23] | Retrospective cross-sectional | Montreal, Canada | 253 households | 1 July 2017–31 March 2021 | Medical records of children (2–17 years) receiving care at a paediatric obesity management clinic | African 13.8% Asian 4.7% Caribbean 13.0% European 4.7% Latin, Central or South American 7.9% Other North American 33.2% Multiple 22.1% | Mean child age 11.3 ± 3.2 years | Before and during pandemic (different individuals) | +validated measure of food security +data routinely collected -small sample receiving care during COVID-19 limiting statistical power |
| Dondi et al [24] | Cross sectional | Italy | 5811 participants | 1st September 2020–15th October 2020 | Parents with children up to 18 years. | - | Number of children: 1 41.4% 2 48.6% 3 7.7% >3 2.3 Age: ≤2 30.3% 3–5 24.4% 6–10 26.2% 11–14 11.8% >14 7.3% | Before the COVID-19 pandemic | +high response rate with 73% complete responses and included in the analysis -convenience sample -retrospective self-report for pre-pandemic data |
| Escobar et al [25] | Cohort (recruited from 3 existing cohorts) | San Francisco Bay Area, USA | 375 households, 1875 individuals | May–September 2020 | Latinx mothers | 100% Latinx | Mean number of children by cohort: TAB: 2.4 ± 1.1 LEAD: 2.7 ± 1.1 HEN: 2.9 ± 1.2 | Before the COVID-19 pandemic | +high risk population group +recruited from existing longitudinal birth cohort studies +validated measure of food security -retention rate not clearly stated, <10% contacted declined but number contacted not stated -results reported separately by cohort |

*(Continued)*

**Table 3.** (Continued)

| Author, year | Study design | Location | Sample size | Data collection period | Population | Ethnicity | Child age | Exposure/ comparison | Quality assessment |
|---|---|---|---|---|---|---|---|---|---|
| Goudie and McIntyre [26] | Cross-sectional studies | 1. Great Britain (GB) 2. GB 3. UK 4. UK 5. UK 6a. UK 6b. UK 7a. UK 7b. UK | 1. 2070 2. 4343 3. 2284 4. 4352 5. 4350 6a. 1064 6b. 10,845 7a. 1308 7b. 4231 participants | 1. 25th-26th March 2020 2. 7th-9th April 2020 3. 24th-29th April 2020 4. 14th-17th May 2020 5. 6th-8th July 2020 6a. 8th-20th Sept 2020 6b. 24th Aug-1st Sept 2020 7a. 22nd Jan-2nd Feb 2021 7b. 29th Jan-2nd Feb 2021 | 1. Adults 18+ 2. Adults 18+ 3. Adults in households with children 4. Adults 18+ 5. Adults 18+ 6a. Children aged 7–17 6b. Adults 18+ 7a. Children aged 7–17 7b. Adults 18+ | - | - | Comparing food insecurity at several timepoints from the beginning of the lockdown restrictions | +nationally representative surveys +children and young people surveyed directly -1 month recall period for food insecurity so prevalence could be lower than a longer recall period (such as 6 months) in surveys 1-6b |
| Kent et al 2022 [27] | Repeated cross-sectional | Tasmania, Australia | Households with children: 373 in May 2020, 271 in Sept 2020, 300 in May 2021 | May 2020, September 2020 and May 2021 | - | - | - | Comparing food insecurity at three timepoints from the beginning of the lockdown restrictions | +data collected at three time-points -convenience sample -inability to link individual responses across the surveys |
| Kowalski et al [28] | Cohort | Maryland, USA | 496 households | Pre-pandemic: Oct 2017 – March 2020. Pandemic: 11th May–11th August 2020 | Caregivers of children involved in the obesity prevention trials. | Non-Hispanic White 51% Non-Hispanic Black or African American 37% Multiracial 5% Hispanic or Latino any race 5% Asian 2% Native American or Alaskan Native 1% Other /did not respond 1% | Child age: 43% 3–5 y 28% 6–10 y 29% 11-15y Number in family: 1 20% 2 42% ≥3 37% | Before the COVID-19 pandemic | +longitudinal data on families before and during the pandemic +/-56% response rate, 47% for responses included in analysis -analysis limited to families eligible for coronavirus aid stimulus payment or who experienced financial shock |

*(Continued)*

**Table 3.** (Continued)

| Author, year | Study design | Location | Sample size | Data collection period | Population | Ethnicity | Child age | Exposure/ comparison | Quality assessment |
|---|---|---|---|---|---|---|---|---|---|
| Lim et al [29] | Cross sectional | USA | 372 participants | May 2020 – February 2021 | 247 adults with cystic fibrosis and 125 primary caregivers of a child with cystic fibrosis | - | - | Before the COVID-19 pandemic | +validated screening measure for food security -convenience sample -participants recruited over 10 month period when COVID-19 restrictions and impact may have varied -retrospective self-report of pre-COVID-19 data |
| MacBain et al 2023 [30] | Cross-sectional | Hamilton, Ontario, Canada | 665 households | Pre-pandemic: 2012 Pandemic: September to December 2021 | Families with a child <18 years | - | - | Before the COVID-19 pandemic | +validated screening measure used for food security +data from 2012 available for comparison -research team had to be present in emergency department to deliver surveys and thus recruitment was subject to availability of research team member |
| Molitor and Doerr [31] | Cohort | California, USA | 11,653 households | Pre-pandemic: 21$^{st}$ Nov 2019–14$^{th}$ March 2020 Pandemic: 27$^{th}$ April–21$^{st}$ July 2020 | Households at or below 185% of the federal poverty line with 1 or more adult women and child(ren) 5–17 years | Latina 65.7% African American 12.6% White 16.8% | - | Before the COVID-19 pandemic | +random sample of eligible households (as part of annual population-based survey) +large sample size -responses missing food security data were excluded |
| Niles et al [32] | Cross sectional with longitudinal subset at one study location | USA | 27,168 participants | March 2020 –February 2021 | Adults, 40.6% in households with children. | Non-Hispanic White 70.0% Non-Hispanic Black 8.1% Hispanic 11.9% Other or multiple 8.1% Not reported 1.4% | - | Before the COVID-19 pandemic | +different sampling techniques across different study sites recruiting convenience, representative and high-risk targeted sample -retrospective report of food insecurity pre-pandemic |

*(Continued)*

**Table 3.** (Continued)

| Author, year | Study design | Location | Sample size | Data collection period | Population | Ethnicity | Child age | Exposure/ comparison | Quality assessment |
|---|---|---|---|---|---|---|---|---|---|
| Parekh et al [33] | Cross sectional | USA | 1452 households | 16[th]– 21[st] April 2020 | Adults, 25.9% living with children <18. | Non-Hispanic White 91.7% Non-White 8.3% | - | Before the COVID-19 pandemic (compared to national prevalence data) | +validated measure for food security +/-assessed food insecurity from start of pandemic instead of pandemic restrictions -convenience sample |
| Sharma et al [34] | Cross sectional | USA | 1048 households | April 2020 | Low-income households with children <18 | Black or African American 7.1% Mexican American, Latino, or Hispanic 85.9% Non-Hispanic White 3.7% Other 3.4% | Mean number of children 2.7 ± 1.1 | Before the COVID-19 pandemic | +targeted low-income families by recruiting through school health program -low response rate (6.4%) |
| St. Pierre et al [35] | Cross sectional | New Orleans, USA | 88 56 participants | October 2020 and April/May 2021 | Students in 6[th] and 7[th] grade (10–14 years) | Black/ African American 94.3% 94.6% Hispanic/ Latino 3.4% 3.6% Multi-racial 2.3% 1.8% | Mean age 11.9 ± 0.8 | Change in food security status over the progression of the COVID-19 pandemic. | +recruited children directly +/-participants recruited to study examining impact of sports-based youth development programme and had to exclude students who were fully virtual due to COVID-19 restrictions as data collection was in person. -small sample size -unvalidated measure of food security |

47% reported food insecurity, 94% of whom indicated that this had begun or worsened during the pandemic [17]. Analysis of data from three waves of the UK Understanding Society Covid Survey showed that 13% of households reported any person in the household being unable to eat healthy and nutritious food and 2% reported being hungry but not eating [21]. A cross-sectional online survey of participants from a longitudinal family-based cohort in April/May 2020 found that 8.5% of mothers and 4.8% of fathers had concerns about food security during the past month and were concerned about food security over the next six months [22].

## Job disruption/loss of income

No studies reported on poverty status explicitly, but there are proxy indicators of possible financial strain in households as it may impact on food insecurity. Overall, seven studies reported on some aspect of pandemic- related job disruption and/or loss of income [18, 19, 21,

**Table 4. Summary of results reported across the included studies.**

| Author, year | Food security measure | Food security/insecurity | Job disruption/reduced income | Mental health | Diet quality |
|---|---|---|---|---|---|
| Abrams et al [17] | 2-question Hunger Vital Sign | 47%<br>94% reported food insecurity begun/worsened during the pandemic | - | - | - |
| Adams et al 2020 [18] | 6 item USDA Household Food Security Module | Food security ↓17% (from 63% to 46.6%)<br>Very low food security ↑20% (from 10 to 30%)<br>Food secure households pre-pandemic:<br>Low food security 15.6%<br>Very low food security 15.3%<br>Low food security pre-pandemic:<br>Very low food security 46.5% | Reduced income 60.1%<br>Job loss or furlough 40.9%<br>Filed for unemployment benefit 34.6% | - | Takeout/fast food ↓62%<br>Home-cooked meals ↑73.3%<br>Total food ↑42% ↓23.5%<br>High calorie snack foods ↑33.2% ↓20%<br>Desserts and sweets ↑35.6% ↓22.8%<br>Fresh foods ↑37% ↓21.8%<br>Non-perishable processed food ↑46.6% ↓13% |
| Adams et al 2021 [19] | 6 item USDA Household Food Security Module | Food security ↓8% from pre-pandemic (from 63%–55%)<br>↑9% from May 2020 (46%–55%)<br>From May 2020, 6.2% became food insecure and 15% became food secure | Reduced income 39.7%<br>Filed for unemployment benefit 19.2% | - | Total food ↑26.6% ↓21%<br>High calorie snack foods ↑17.3% ↓29.3%<br>Desserts and sweets ↑15.5% ↓32.8%<br>Fresh foods ↑36.3% ↓17.3%<br>Non-perishable processed food ↑33.3% ↓17.3% |
| Adams et al 2023 [20] | 6 item USDA Household Food Security Module | Food security back to pre-pandemic levels (from 65.2%–64.6%)<br>Very low food security ↑7% from pre-pandemic (9.6%–16.8%) | - | - | - |
| Brown et al [21] | 2 questions based on FAO definition | Any person in household unable to eat healthy and nutritious food because of money or other resources 13%<br>Hungry but did not eat 2% | Newly unemployed 8%<br>Furloughed 11% | - | - |
| Carroll et al [22] | Food insecurity concern by adapting 1 question from Hunger Vital Sign | Food security concerns in the past month and the next 6 months 8.5% mothers, 4.8% fathers | Financial stress in the past month 19% mothers, 14% fathers<br>Financial stress in the next six months 22% mothers, 18% fathers | Parents reported moderately high levels of stress (mothers 6.8, fathers 6.0)<br>Financial stress (19% m, 14% f) | Total food ↑57% mothers, 46% fathers, 42% children<br>Snack food ↑67% mothers, 59% fathers, 55% children<br>Fast food/takeout ↓43% mothers, 45% father |
| Cyrenne-Dussault et al 2022 [23] | Health Canada's 18-item Household Food Security Survey Module (HFSSM) | Household food insecurity ↑5.5 from before (39.6%) to during (45.1%) pandemic<br>Child food insecurity ↑1.9 from before (20.7%) to during (22.6%) pandemic | - | - | - |

*(Continued)*

**Table 4.** (Continued)

| Author, year | Food security measure | Food security/insecurity | Job disruption/reduced income | Mental health | Diet quality |
|---|---|---|---|---|---|
| Dondi et al [24] | 2-question Hunger Vital Sign | Worried about running out of food:<br>↑5.6 sometimes true<br>↑2.3% often true<br>Ran out of food and did not have money to buy more:<br>↑0.8% sometimes true<br>↑0.5% often true | Furloughed 38.3%<br>Job loss 4.4% | - | Total food ↑27.3% of which<br>Snacks ↑60.3%<br>Fruit juices ↑14%<br>Soft drinks ↑10.4% |
| Escobar et al [25] | 18 item USDA Household Food Security Module | Household food security by recruitment cohort:<br>LEAD: ↓47.7% compared to 1 year ago<br>HEN: ↓44.8% compared to subsample six months prior to COVID<br>HEN: ↓22.4% compared to >5 years ago<br>Child food security by recruitment cohort:<br>LEAD: ↓31% compared to 1 year ago<br>HEN: ↓28.8% compared to subsample six months prior to COVID<br>HEN: ↓15.2% compared to >5 years ago | - | - | - |
| Goudie and McIntyre [26] | 3 modified items from the 10-item USDA Adult Food Security Module for adults and four questions for child insecurity also used by the USDA | Food insecurity (1 month recall period) in households with children:<br>March 2020: 20.8%<br>May 2020: 12.4%<br>July 2020: 12.6%<br>August 2020: 10.8%<br>January 2021 9.6%<br>July 2020-January 2021: 12%<br>↓11.2% over recording period<br>Food insecurity over summer holiday in children 8–17 years 18%<br>Food security in households with ≥3 children ↑4.2% compared to pre-Covid | - | - | - |
| Kent et al 2023 [27] | 6 item USDA Household Food Security Module | Food insecurity in households with children:<br>May 2020: 29.8%<br>September 2020: 22.1% (↓7.6% from May 2020)<br>May 2021: 24.7% (↓5.1% from May 2020) | - | - | - |
| Kowalski et al [28] | 2-question Hunger Vital Sign | Food insecurity ↑3% (from 22% to 25%) | Early pandemic change in household income ↑3% ↓39%<br>Reduced hours 20%<br>Temporary or permanent job loss 20% | - | - |

*(Continued)*

**Table 4.** (Continued)

| Author, year | Food security measure | Food security/insecurity | Job disruption/reduced income | Mental health | Diet quality |
|---|---|---|---|---|---|
| Lim et al [29] | 2-question Hunger Vital Sign | Food insecurity ↑4% in children (from 23.3% to 27.2%) ↑2.4% in adults (from 16.6% to 19.0%) | - | Depression ↑4.1% (from 13.4% to 17.5%) Anxiety ↑4.3% (from 17.7% to 22.0%) Abnormal health screen ↑4.6% (from 19.6% to 24.2%) | - |
| MacBain et al 2023 [30] | 2-question Hunger Vital Sign | Food insecurity ↑3.3% from 2012 (22.7%) to 2021 (26%) in families presenting to the emergency department with a paediatric patient Food insecurity ↑10% in study population than the Canadian (15.8%) and Ontario (16.1%) average for 2020 | - | - | - |
| Molitor and Doerr [31] | 6-item USDA Household Food Security Module | Very low food security: 14% post COVID-19 ↓5% in 2020 post COVID-19 compared to 2018 ↓8.2% in 2020 post COVID-19 compared to 2019 ↓5.3% in 2020 post COVID-19 compared to 2020 pre COVID-19 | - | - | - |
| Niles et al [32] | 6-item USDA Household Food Security Module | Food insecurity: Convenience sample ↑8.9% (30.1%–39%) State representative ↑12% (37.2%–49.2%) High risk ↑13.5% (44.1%–57.6%) | Job disruption/reduced income 35.3% | - | - |
| Parekh et al [33] | 6-item USDA Household Food Security Module | Low food security 10.3% Very low food security 7.2% ↑3.9% compared to 2019 national data (13.6%) | - | - | - |
| Sharma et al [34] | 2-question Hunger Vital Sign | Food insecurity ↑22% 71.5% in 2019 to 93.5% in 2020 | - | - | - |
| St. Pierre et al [35] | One question on whether the person would "worry about the amount or type of food available to you at home due to money or lack of availability" (yes/no)) | Food insecurity ↑1.9% 28.4% in October 2020 to 30.3% in April/May 2021 | - | - | Low fruit and vegetable intake: 68.2% in October 2020 and 74.1% in April/May 2021 High sugar sweetened beverage intake: 68.6% in October 2020 and 54.5% in April/May 2021 |

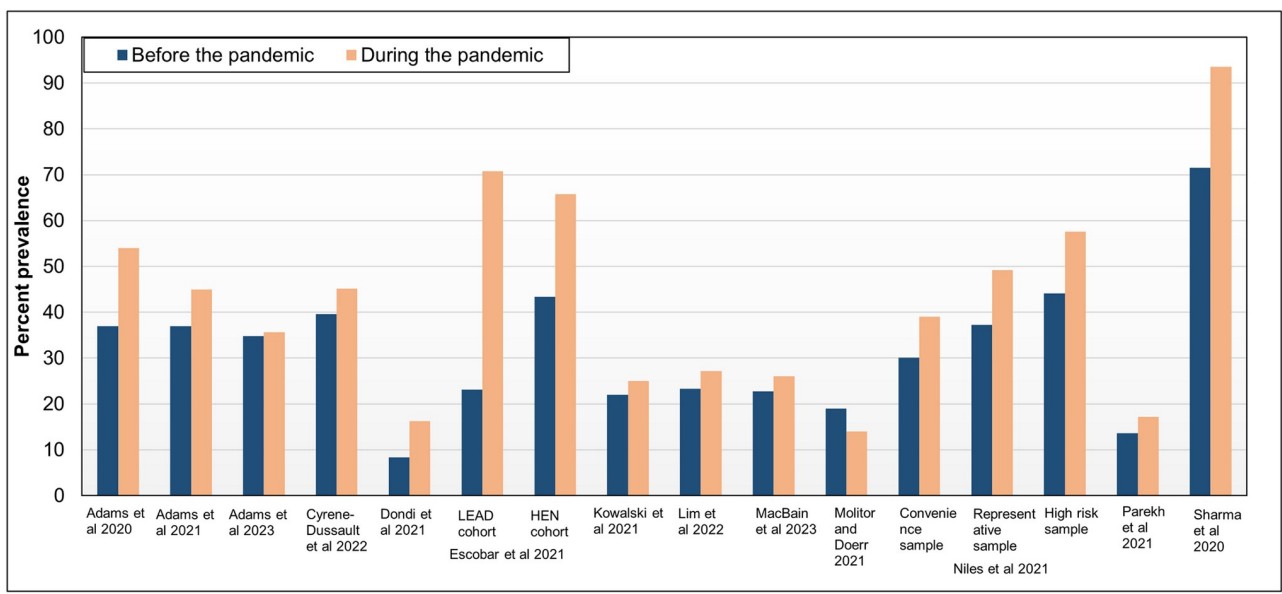

**Fig 2. The prevalence of food insecurity in the included studies before and during the pandemic.**

24, 26, 28, 32] and one study provided an indication of financial stress [22]. Four of these studies reported on job loss and/or furlough (suspended from employment with pay). Three studies reported similar proportions respondents being affected by factors related to job disruption or loss of income (40% [28], 40.9% [18] and 42.7% [24]). The proportion of the population affected by job/income disruption was lower in the fourth study (18%) [21].

Four studies reported on decrease in income during the pandemic in the study population [18, 19, 26, 28]. This ranged from 22% who reported a drop in income across the course of the pandemic (March 2020 to January 2021) [26] to 60.1% who lost income early in the pandemic (April/May 2020) [18]. However, on follow-up, the proportion who reported a recent decrease in income had reduced to 39.7% in September 2020 [19]. Finally, one study reported a combined percentage of 35.3% who experienced job disruption or decrease in income [32].

In the study that reported on financial stress [22], 19% of mothers and 14% of fathers reported financial stress in the past month with a higher proportion expecting to experience financial stress over the next six months (22% of mothers and 18% of fathers).

Prevalence of food insecurity was higher among participants facing job disruption/loss of income [18, 32]. Respondents who reported financial issues only as a reason for food insecurity over the course of the pandemic experienced an increase in food insecurity from 16% in the first two weeks of lockdown (March 2020) to 26% in May 2020 and 42% in January 2021 [26]. Families who were food secure pre-pandemic that faced decreased income, job loss or reduction in working hours were more than twice as likely to be at risk of early pandemic food insecurity [28] as were furloughed participants [21].

## Mental health

Two studies considered mental health outcomes [22, 29]. One study [22] asked parents to self-report stress by rating on a scale from 1 (no stress) to 10 (extreme stress) and found that parents reported moderately high levels of stress, with an average score of 6.0 (standard deviation (SD) 2.5) in fathers and 6.8 (SD 1.9) in mothers. Parents were also asked to report on their

child's stress about COVID-19 with a three-item scale with 45% reporting their child was "somewhat concerned" or "very concerned". The other study [29] assessed mental health using the Patient Health Questionnaire-4 (two questions each on anxiety and depression) and found an increase in depression (17.5% during the pandemic, 13.4% before the pandemic), anxiety (22.0% during, 17.7% before) and abnormal mental health screen (24.2% during 19.6% before) in their sample compared to before the pandemic. Food insecure participants were found to be more likely to have depression and anxiety than food insecure participants, both before and during the pandemic. Both food secure and insecure participants reported worsening of mental health during the pandemic but the change in prevalence was more in food insecure participants.

### Diet quality

Five studies included some aspect of diet quality [18, 19, 22, 24, 35]. Two studies assessed the home food environment [18, 19], two studies reported on change in diet behaviours [22, 24] and one study assessed fruit and vegetable and sugar sweetened beverage intake [35].

Two studies assessed the family's home food environment at two time points by asking questions about the availability of high calorie snack food, desserts and sweets, fresh foods, non-perishable processed food and total amount of food [18, 19]. Total amount of food increased for 56% of food secure families, but decreased for 53% of families who experienced very low food security. Non-perishable, processed food items increased the most in households with very low food security during the pandemic (55.9%), compared to an increase of 46.0% in households experiencing low food security and 40.6% in food secure households. Fresh foods in the home increased by 36% of food secure households and 38% in food insecure households, but decreased in households with low food security (24%) and in households with very low food security (36%) [18]. The follow-up survey in September 2020 [19] found that about half of families did not change the amount of different types of food available. Among families that reported change, a greater percentage decreased the amount of high-calorie snack food and desserts and sweets, and a greater percentage increased the amount of fresh foods and non-perishable processed foods [19].

In one of the studies that assessed diet behaviours [22], parents self-reported their own and their child's change in eating. Most parents reported change (70% of mothers, 60% fathers, 51% children). The most common changes in children were eating more food of any type (42%) and eating more snack food (55%). In the other study [24], parents reported their child's change in eating with 27.3% eating more food of any type. Parents specifically reported an increase in consumption of snacks (60.3%), fruit juice (14%) and soft drinks (10.4%).

One further study [35] asked participants (students aged 10–14 years) to indicate portions of fruit and vegetables and sugar sweetened beverages consumed per day in the previous two weeks. Less than a third met the 5-a-day recommendation for fruit and vegetables and about 30% reported consuming less than one portion per day. Over 80% reported consuming one or more sugar sweetened beverages per day [35].

### Weight status

No studies assessed weight change.

## Discussion

This systematic review included 19 studies that examined the impact of the COVID-19 pandemic on food security in households with children. Most studies that compared prevalence of food insecurity before and during the COVID-19 pandemic found that food insecurity

increased in households with children. A high proportion of participants reported job and income losses due to the COVID-19 pandemic. One study [31] reported a decrease in food insecurity likely due to pandemic responsive actions by the government as part of which nutritional assistance benefits increased. This review builds on existing evidence exploring the impact of the COVID-19 pandemic on food insecurity. A review of nine studies on the effects of the pandemic in Australia found an increase of food insecurity in independent participants of all ages due to multiple economic and physical barriers due to the COVID-19 pandemic [38]. These included the government travel restrictions and consumer stockpiling, which reduced the ability of people to access sufficient food of the right nutritional balance for a healthy diet [38]. Our review expands on these findings by widening the range of countries included and by focusing on households with children as our population.

The findings on mental health in this review complement the findings of a review of 28 studies about mental health issues linked to COVID-19 which found an increase in symptoms of depression, stress and anxiety attributed to quarantine and disruption, economic worries, and fear of illness [39].

Although our findings suggest that diet composition had changed for households with children during the COVID-19 pandemic, this conflicts with a systematic review of 38 studies which concluded there was insufficient evidence to infer changes in diet quality during the pandemic, however, only four studies included children in their study population [40]. Our review shows that food insecurity in households with children increased during the pandemic. The studies that examined the prevalence during the course of the pandemic found that the prevalence of food insecurity was highest when restrictions were first implemented and improved during the course of the pandemic when restrictions were eased. The prevalence of very low food security remained high later in the pandemic even though the prevalence of food security had returned to pre-pandemic levels [20].

Future research should focus on how food insecurity changed over the course of the pandemic, impact of the discontinuation of pandemic responsive actions and to identify adaptations to current support schemes to address food insecurity. Research into the long-term impact of the COVID-19 pandemic on food insecurity in households with children is recommended to see if the effects are maintained, worsened, or improved; and to identify factors associated with any changes. Strategies to address the risk factors and improve the long-term prognosis of those with food insecurity could involve government organisations mandating a liveable wage in proportion to inflation and cost of living increase.

The strengths of this review include a comprehensive search strategy, using a range of electronic databases as well as a search of the grey literature surrounding the topic, to collate as many studies as possible. Use of a second reviewer for both screening and quality assessment is a strength as this reduces risk of bias in this review. However, limitations of this review include excluding studies that were not published in English, as this may lead to relevant literature not being included. This is particularly relevant as our review included countries from the OECD [13], and many of these countries do not have English as their primary language. This review looks at populations in countries in the OECD. However, nearly two-thirds (63.2%) of the studies included in this review, were based in the USA [17–20, 25, 28, 29, 31–35]. This impacts the generalisability of our findings due to the lack of diversity in countries. Additionally, lack of diversity in recruited population groups and oversampling of high-risk groups leads to a non-representative sample, also limiting the generalisability. Multiplicity of the population included in each study made it difficult for comparison. Although the specified population for this review was households with children under 18 years, some studies used a smaller age group or households as their population, with children as a subgroup. Over two-thirds of the studies included in this review were cross-sectional and relied on retrospective report of

food insecurity prior to the pandemic. Although studies used different measures for food insecurity, the measures used were commonly used in research and/or practice with the exception of three studies which adapted questions to capture aspects of food insecurity. Other outcomes assessed in this review were measured through study specific questions and thus were not comparable across different studies.

In summary, this review found that the COVID-19 pandemic was associated with worsening of food security in households with children. Increase in stress and worsening of mental health outcomes during the pandemic was reported in the studies that examined these outcomes. Schemes that improve food access could be protective against food insecurity but high rates of food insecurity were found in those accessing food benefit schemes highlighting the need to review the level of support provided by these schemes. Protection against food insecurity should be factored in pandemic preparation.

## Supporting information

**S1 Checklist. PRISMA 2020 for abstracts checklist.**
(DOCX)

**S2 Checklist. PRISMA 2020 checklist.**
(DOCX)

## Author Contributions

**Conceptualization:** Anna Williams, Nisreen A. Alwan, Dianna Smith, Nida Ziauddeen.

**Data curation:** Anna Williams, Elizabeth Taylor, Nida Ziauddeen.

**Formal analysis:** Anna Williams, Nida Ziauddeen.

**Funding acquisition:** Nisreen A. Alwan, Dianna Smith, Nida Ziauddeen.

**Methodology:** Nisreen A. Alwan, Elizabeth Taylor, Nida Ziauddeen.

**Project administration:** Anna Williams.

**Supervision:** Nisreen A. Alwan.

**Writing – original draft:** Anna Williams.

**Writing – review & editing:** Nisreen A. Alwan, Elizabeth Taylor, Dianna Smith, Nida Ziauddeen.

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
