## [Decision Letter · Decision Letter 0]

2 Jul 2024

PONE-D-24-06499The COVID-19 pandemic and food insecurity in households with children: a systematic reviewPLOS ONE

Dear Dr. Ziauddeen,

Thank you for submitting your manuscript to PLOS ONE. After careful consideration, we feel that it has merit but does not fully meet PLOS ONE’s publication criteria as it currently stands. Therefore, we invite you to submit a revised version of the manuscript that addresses the points raised during the review process.

We look forward to receiving your revised manuscript.

Kind regards,

George N Chidimbah Munthali

Academic Editor

PLOS ONE

“The Wessex DIET study which this review formed part of is supported by the NIHR Applied Research Collaboration Wessex. The funder had no role in study design, data collection and analysis, decision to publish, or preparation of the manuscript. The views and opinions expressed in this protocol are those of the authors and do not necessarily reflect those of the NIHR or the Department of Health and Social Care. For the purpose of Open Access, the author has applied a Creative Commons Attribution (CC BY) licence to any Author Accepted Manuscript version arising.”

3. We noted in your submission details that a portion of your manuscript may have been presented or published elsewhere. [An abstract of this work was accepted for a poster presentation at the Society for Social Medicine and Population Health Annual Scientific Meeting held in September 2023 and the submitted abstract was published in a supplement of the Journal of Epidemiology and Community Health (https://jech.bmj.com/content/77/Suppl_1/A78.1).] Please clarify whether this [conference proceeding or publication] was peer-reviewed and formally published. If this work was previously peer-reviewed and published, in the cover letter please provide the reason that this work does not constitute dual publication and should be included in the current manuscript.

Additional Editor Comments:

Dear Dr Nida Ziaddeen

I would like to inform you that your manuscript has undergone the first review process. Kindly would you address the comments by the reviewers and submit the revised manuscript with point to point answer.

Regards

George Munthali

Reviewers' comments:

Reviewer's Responses to Questions

**Comments to the Author**

1. Is the manuscript technically sound, and do the data support the conclusions?

Reviewer #1: Yes

Reviewer #2: Yes

2. Has the statistical analysis been performed appropriately and rigorously? 

Reviewer #1: Yes

Reviewer #2: N/A

3. Have the authors made all data underlying the findings in their manuscript fully available?

Reviewer #1: Yes

Reviewer #2: Yes

4. Is the manuscript presented in an intelligible fashion and written in standard English?

Reviewer #1: Yes

Reviewer #2: Yes

5. Review Comments to the Author

Reviewer #1: This is a well researched and written paper. I found the methodology and results presentation well articulated. It would be interesting to look at the nutritional status in developing countries considering their vulnerability.

Reviewer #2: Comments

The manuscript presents a thorough systematic review of the impact of the COVID-19 pandemic on food insecurity in households with children in high-income countries. The review is well-structured, and the methodology appears rigorous. However, the inclusion criteria and exclusion criteria need to be explicitly stated to avoid any ambiguity. The conclusions drawn are generally supported by the data, but the narrative synthesis could benefit from a more detailed comparison of the methodologies and findings across the included studies.

Statistical Analysis

While the manuscript does not include original statistical analysis, it relies on the statistical results of the included studies. The use of the National Heart, Lung, and Blood Institute (NIH) tool for quality assessment is appropriate. However, the manuscript should provide more detailed information on the statistical rigor of the included studies, such as sample sizes, control for confounding factors, and the robustness of their statistical methods. This will help in assessing the overall reliability of the review’s conclusions.

The manuscript is generally well-written and presented in standard English. However, there are several instances of typographical and grammatical errors that need correction. For example, the use of symbols like "�" and "�" should be looed into. Additionally, some sentences are overly complex and could be simplified for better readability. Ensuring consistency in terminology and improving sentence structure will make the manuscript more accessible.

Review

General comments on the sections

Introduction: The introduction provides a good overview of the background and significance of the study. However, it would benefit from a more detailed explanation of the conceptual framework guiding the review, particularly how the PICO framework was operationalized.

Methods: The search strategy is comprehensive, but the inclusion and exclusion criteria should be more explicitly detailed. Additionally, the authors should provide more information on the process of quality assessment and how disagreements between reviewers were resolved.

Results: The results are well-organized, but the presentation could be enhanced by including more detailed tables summarizing the characteristics and findings of the included studies. A meta-analysis, if feasible, could provide more robust quantitative synthesis.

Discussion: The discussion appropriately contextualizes the findings within the broader literature. However, it should include a more critical analysis of the limitations of the included studies and the review itself. Suggestions for future research should be more specific, focusing on gaps identified in the current literature.

Conclusion: The conclusion effectively summarizes the key findings but should be more concise. Emphasizing the practical implications of the findings for policymakers and practitioners would strengthen the manuscript.

The manuscript’s strength lies in its comprehensive search strategy and inclusion of grey literature. However, the lack of diversity in the included studies, predominantly from the USA, limits the generalizability of the findings. The authors should consider this limitation and discuss its implications more thoroughly.

The narrative review format is appropriate given the heterogeneity of the included studies, but a more detailed exploration of the methodological differences and their potential impact on the findings would be beneficial.

Ethical considerations related to the inclusion of studies involving human participants should be explicitly addressed in the methods section.

Overall, the manuscript addresses an importanttopic, but it requires some revisions to improve clarity, rigor, and comprehensiveness.

6. PLOS authors have the option to publish the peer review history of their article (what does this mean?). If published, this will include your full peer review and any attached files.

Reviewer #1: No

Reviewer #2: No

---

## [Author Response · Author response to Decision Letter 0]

27 Jul 2024

Additional requirements:

Thank you, we have followed the PLOS ONE style requirements for the revised version. 

“The Wessex DIET study which this review formed part of is supported by the NIHR Applied Research Collaboration Wessex. The funder had no role in study design, data collection and analysis, decision to publish, or preparation of the manuscript. The views and opinions expressed in this protocol are those of the authors and do not necessarily reflect those of the NIHR or the Department of Health and Social Care. For the purpose of Open Access, the author has applied a Creative Commons Attribution (CC BY) licence to any Author Accepted Manuscript version arising.”

EJT and NZ were 

Thank you, we have included the amended statement within our cover letter and the funding section in the manuscript.

3. We noted in your submission details that a portion of your manuscript may have been presented or published elsewhere. [An abstract of this work was accepted for a poster presentation at the Society for Social Medicine and Population Health Annual Scientific Meeting held in September 2023 and the submitted abstract was published in a supplement of the Journal of Epidemiology and Community Health (https://jech.bmj.com/content/77/Suppl_1/A78.1).] Please clarify whether this [conference proceeding or publication] was peer-reviewed and formally published. If this work was previously peer-reviewed and published, in the cover letter please provide the reason that this work does not constitute dual publication and should be included in the current manuscript.

The abstract was reviewed by the conference organising committee and scored (score not provided to authors) for decision making on acceptance of abstract to the conference programme. The authors received no peer review comments and the abstract was published as submitted in a supplement of the Journal of Epidemiology and Community Health which is the usual practice for this conference accepted abstracts every year. This does not constitute dual publication as the conference abstract does not include the full methods, results and discussion included in the paper. Also the published abstract only includes the initial search and does not include the updated search. 

Thank you, we have added the suggested sentence for research that concerns only data provided in our submission. 

The supporting information included are the PRISMA checklists required as part of the submission. There are no captions or in-text citations associated with this. 

Thank you, we have reviewed the reference list as suggested. 

Additional Editor Comments:

I would like to inform you that your manuscript has undergone the first review process. Kindly would you address the comments by the reviewers and submit the revised manuscript with point to point answer.

Thank you, point to point answer to reviewers comments included below. 

Reviewers' comments:

Reviewer #1: This is a well researched and written paper. I found the methodology and results presentation well articulated. It would be interesting to look at the nutritional status in developing countries considering their vulnerability.

Thank you for the feedback. We agree that it would be interesting to look at the status in developing countries but this was out of the scope of this review. 

Reviewer #2: Comments

The manuscript presents a thorough systematic review of the impact of the COVID-19 pandemic on food insecurity in households with children in high-income countries. The review is well-structured, and the methodology appears rigorous. However, the inclusion criteria and exclusion criteria need to be explicitly stated to avoid any ambiguity. The conclusions drawn are generally supported by the data, but the narrative synthesis could benefit from a more detailed comparison of the methodologies and findings across the included studies.

Statistical Analysis

While the manuscript does not include original statistical analysis, it relies on the statistical results of the included studies. The use of the National Heart, Lung, and Blood Institute (NIH) tool for quality assessment is appropriate. However, the manuscript should provide more detailed information on the statistical rigor of the included studies, such as sample sizes, control for confounding factors, and the robustness of their statistical methods. This will help in assessing the overall reliability of the review’s conclusions.

The manuscript is generally well-written and presented in standard English. However, there are several instances of typographical and grammatical errors that need correction. For example, the use of symbols like "�" and "�" should be looed into. Additionally, some sentences are overly complex and could be simplified for better readability. Ensuring consistency in terminology and improving sentence structure will make the manuscript more accessible.

General comments on the sections

Introduction: The introduction provides a good overview of the background and significance of the study. However, it would benefit from a more detailed explanation of the conceptual framework guiding the review, particularly how the PICO framework was operationalized.

We used the following figure to conceptualise factors due to COVID-19 that may have impacted prevalence of food insecurity. However, we did not include this in the manuscript as did not think it was needed. We are happy to include as a supplementary figure if the reviewer thinks it would be relevant to do so. 

We are not entirely sure what the reviewer means by operationalising the PICO framework. We used the PICO framework to identify the population, intervention and control group and the outcome of interest to develop the research question. 

Methods: The search strategy is comprehensive, but the inclusion and exclusion criteria should be more explicitly detailed. Additionally, the authors should provide more information on the process of quality assessment and how disagreements between reviewers were resolved.

Thank you, we have added Table 1 to explicitly detail the inclusion and exclusion criteria. 

We have included details of the quality assessment in the methods section. Both reviewers independently assessed quality of each study and then resolved any disagreements through discussion. 

Results: The results are well-organized, but the presentation could be enhanced by including more detailed tables summarizing the characteristics and findings of the included studies. A meta-analysis, if feasible, could provide more robust quantitative synthesis.

Thank you, we have included the characteristics and findings of the included studies in Tables 3 and 4 and Fig 2. We believe we have captured all the relevant information in these tables but are happy to look into adding additional information (if available) if the reviewer could clarify the further details that would be helpful to include. 

A meta-analysis was not feasible due to the heterogeneity of the included studies and thus was not included in the analysis. 

Discussion: The discussion appropriately contextualizes the findings within the broader literature. However, it should include a more critical analysis of the limitations of the included studies and the review itself. Suggestions for future research should be more specific, focusing on gaps identified in the current literature.

Thank you, we have added a paragraph on limitations of the included studies. Limitations of the review itself are considered in the discussion section. The fourth paragraph in the discussion focused on suggestions for future research which are based on the gaps identified in the literature and we have added two further points. 

Conclusion: The conclusion effectively summarizes the key findings but should be more concise. Emphasizing the practical implications of the findings for policymakers and practitioners would strengthen the manuscript.

Thank you, we have added wording around the practical implications of the findings. 

The manuscript’s strength lies in its comprehensive search strategy and inclusion of grey literature. However, the lack of diversity in the included studies, predominantly from the USA, limits the generalizability of the findings. The authors should consider this limitation and discuss its implications more thoroughly.

We have considered these in the limitations section. 

The narrative review format is appropriate given the heterogeneity of the included studies, but a more detailed exploration of the methodological differences and their potential impact on the findings would be beneficial.

Ethical considerations related to the inclusion of studies involving human participants should be explicitly addressed in the methods section.

Thank you, we have not addressed ethical consideration in the methods section as all the data included in this review was published and thus the review itself did not require ethical approval. 

Overall, the manuscript addresses an important topic, but it requires some revisions to improve clarity, rigor, and comprehensiveness.

Thank you, we have revised and provided responses to the comments provided.

---

## [Editor Report · Decision Letter 1]

30 Jul 2024

The COVID-19 pandemic and food insecurity in households with children: a systematic review

PONE-D-24-06499R1

Dear Dr. Nida Ziauddeen, PhD

We’re pleased to inform you that your manuscript has been judged scientifically suitable for publication and will be formally accepted for publication once it meets all outstanding technical requirements.

Kind regards,

George N Chidimbah Munthali

Academic Editor

PLOS ONE

---

## [Editor Report · Acceptance letter]

31 Jul 2024

PONE-D-24-06499R1 

PLOS ONE

Dear Dr. Ziauddeen, 

I'm pleased to inform you that your manuscript has been deemed suitable for publication in PLOS ONE. Congratulations! Your manuscript is now being handed over to our production team.

Kind regards, 

on behalf of

Mr George N Chidimbah Munthali 

Academic Editor

PLOS ONE